# Developing the Breast Utility Instrument, a preference-based instrument to measure health-related quality of life in women with breast cancer: Confirmatory factor analysis of the EORTC QLQ-C30 and BR45 to establish dimensions

Teresa C. O. Tsui[1,2]*, Maureen Trudeau[3], Nicholas Mitsakakis[4,5], Sofia Torres[6], Karen E. Bremner[1], Doyoung Kim[7], Aileen M. Davis[6‡], Murray D. Krahn[1,2,6‡]

**1** Toronto Health Economics and Technology Assessment (THETA) Collaborative, University Health Network, Toronto, Ontario, Canada, **2** Graduate Department of Pharmaceutical Sciences, Leslie Dan Faculty of Pharmacy, University of Toronto, Toronto, Ontario, Canada, **3** Odette Cancer Centre, Sunnybrook Health Sciences Centre, Toronto, Ontario, Canada, **4** Children's Hospital of Eastern Ontario, Ottawa, Ontario, Canada, **5** Division of Biostatistics, Dalla Lana School of Public Health, University of Toronto, Toronto, Ontario, Canada, **6** Institute of Health Policy, Management and Evaluation, Dalla Lana School of Public Health, University of Toronto, Toronto, Ontario, Canada, **7** Department of Pharmacology and Toxicology, Temerty Faculty of Medicine, University of Toronto, Toronto, Ontario, Canada

‡ AMD and MDK are joint senior authors on this work.
* teresa.tsui@utoronto.ca

## Abstract

### Objectives

Breast cancer (BrC) and its treatments impair health-related quality of life (HRQoL). Utility is a measure of HRQoL that includes preferences for health outcomes, used in treatment decision-making. Generic preference-based instruments lack BrC-specific concerns, indicating the need for a BrC-specific preference-based instrument. Our objective was to determine dimensions of the European Organisation for Research and Treatment of Cancer (EORTC) general cancer (QLQ-C30) and breast module (BR45) instruments, the first step in our development of the novel Breast Utility Instrument (BUI).

### Methods

Patients (n = 408) attending outpatient BrC clinics at an urban cancer centre, and representing a spectrum of BrC health states, completed the QLQ-C30 and BR45. We performed confirmatory factor analysis of the combined QLQ-C30 and BR45 using mean-and variance-adjusted unweighted least squares estimation. The hypothesized factor model was based on clinical relevance, item distributions, missing data, item-importance, and internal reliability of dimensions. Models were evaluated based on global and item fit, local areas of strain, and likelihood ratio tests of nested models.

**Data Availability Statement:** Data cannot be shared publicly because data access is restricted by patient consent to use their study data. Anonymized data are available and can be obtained with approval of the Chair of the Sunnybrook Research Institute Research Ethics Board (current contact information available at: https:// sunnybrook.ca/research/content/?page=sri-crs-reo-home, or telephone: 1-416-480-6100, ext. 88144) and the Chair of the University Health Network Research Ethics Board (current contact information available at: https://www.uhnresearch. ca/content/contacts1, or email: reb@uhnresearch. ca, telephone: 1- 416-581-7849), and approved institutional data sharing agreements. In Canada, research data are the property of the institution, not the investigators. Request for the data may be sent to the corresponding author (teresa.tsui@utoronto. ca).

**Funding:** The authors received financial support for patient honoraria from the THETA Fund for Excellence (# 5790 6839 0706), and for the Biomatrix data warehouse from Dr. Kathleen Pritchard's direct donation.

**Competing interests:** The authors have declared that no competing interests exist.

## Results

Our final model had 10 dimensions: physical and role functioning, emotional functioning, social functioning, body image, pain, fatigue, systemic therapy side effects, sexual functioning and enjoyment, arm and breast symptoms, and endocrine therapy symptoms. Good overall model fit was achieved: $\chi^2$/df: 1.45, Tucker-Lewis index: 0.946, comparative fit index: 0.951, standardized root-mean-square residual: 0.069, root-mean-square error of approximation: 0.033 (0.030–0.037). All items had salient factor loadings ($\lambda$>0.4, p<0.001).

## Conclusions

We identified important BrC HRQoL dimensions to develop the BUI, a BrC-specific preference-based instrument.

## Introduction

Breast cancer (BrC) is the most common cancer, diagnosed in one in eight women during her lifetime [1], with one of the highest per-patient health system costs [2]. Treatments have increased progression-free and overall survival [3–5], however, health-related quality of life (HRQoL) is another important outcome in BrC [6]. Health utility is a preference-based measure of HRQoL, anchored at 0 (dead) and 1 (full health). Utility multiplied by length of life produces quality adjusted life years (QALYs), a key outcome in cost-utility analyses [7].

Existing methods to measure health utilities in BrC have limitations. Notably, generic (e.g. EQ-5D) [8], and general cancer (e.g., e.g., QLU-C10D, EORTC-8D) [9] utility instruments lack construct validity in key BrC-specific dimensions such as arm and breast symptoms, endocrine therapy symptoms, and endocrine sexual symptoms. A BrC-specific preference-based instrument may discriminate better among different BrC health states and be more responsive to mild disease-specific changes in BrC HRQoL than generic instruments [10–12], allowing cost-utility analyses to integrate data derived from more comprehensive, and more valid, health utility measurement [11]. Therefore, our overall objective is to develop the novel Breast Utility Instrument (BUI), a BrC-specific preference-based instrument.

Our overall research program aims to develop and validate the EORTC-derived BrC-specific preference-based instrument, the Breast Utility Instrument (BUI). Novel preference-based instruments are frequently derived from existing psychometric instruments which contain key disease-specific dimensions [12, 13]. Building on Brazier et al's stages of deriving a preference-based HRQoL instrument (12), we developed a 17-step framework, spanning four phases of instrument development: i) develop initial questionnaire items, ii) establish dimension structure, iii) reduce items per dimension, iv) value and model health state utilities (unpublished). The specific objective of this study was to identify dimensions that might be used in developing the BUI, a BrC-specific utility instrument by performing confirmatory factor analysis on the European Organisation for Research and Treatment of Cancer (EORTC) general cancer (QLQ-C30) and breast-specific module (BR45) instruments [14].

## Methods

### EORTC QLQ-C30 and BR45 instruments

The QLQ-C30 version 3 developed in 1993 [15] is a 30-item general cancer HRQoL patient-reported instrument with subscales representing functioning (physical, role, emotional,

cognitive, social), symptoms (fatigue, nausea and vomiting, pain, dyspnea, insomnia, appetite loss, constipation, diarrhea, financial difficulties), and global health items. Each subscale has multiple items, except for six single-item symptom subscales. Each item has four response categories from 1 "not at all" to 4 "very much". The global health items are rated from 1 "very poor" to 7 "excellent" [15].

The QLQ-C30 has demonstrated measurement properties in a range of cancers including breast cancer [16]. It has an established factor structure (construct validity) consistent with the original development population in lung cancer. Its internal consistency assessed by Cronbach's alpha was >0.7 for all subscales except for role functioning and cognitive functioning where alpha was <0.70. Discrimination between local-regional and metastatic BrC was demonstrated in 6/9 subscales at pre-treatment (p<0.002) and in 4/9 subscales (p<0.002) 8 days after chemotherapy [16]. Comparing local-regional and metastatic BrC, subscales *without* significant difference in mean scores pre-treatment were: emotional functioning, cognitive functioning, and nausea / vomiting, and subscales *without* significant differences 8 days after chemotherapy were: emotional functioning, social functioning, cognitive functioning, nausea and vomiting, and fatigue [16]. The QLQ-C30 has established patient-observer agreement with a median kappa = 0.5 (range: 0.49–1.00) in patients with breast and gynecological cancers [17].

The BR45 is a BrC-specific module [14], updated in 2020 from the BR-23 originally developed in 1996 [18] with new (*italicized*) functioning and symptom scales to reflect current treatments [14]. The BR45 has five functioning sub-scales (body image, future perspective, sexual functioning, sexual enjoyment, *breast satisfaction*), and seven symptom subscales (arm, breast, *endocrine therapy*, *skin mucositis*, *endocrine sexual symptoms*, systemic therapy side effects, and upset by hair loss). It also has three open-ended items to capture additional symptoms or problems not addressed by the previous items. All BR45 items have the same four response options as the QLQ-C30 [14].

The developers of the BR45 pre-tested the breast module to evaluate the importance, comprehensibility, and acceptability of its questionnaire items (face validity and feasibility) [14]. The BR45 has also established preliminary psychometric properties, where all subscales have acceptable internal consistency (Cronbach's alpha > 0.7), and the three new symptom subscales and new satisfaction subscale had no strong correlation with the existing BR23 subscales [14].

## Participants and procedures

**Patients.** Between September 2018 and August 2019, a cross-sectional sample of 1,536 patients diagnosed with invasive BrC were screened using appointment lists of six medical oncologists' clinics and an electronic chart review at an urban hospital-based outpatient breast cancer centre. We identified 1,260 potentially eligible patients who were approached in clinic (S1 Fig). Of the 703 patients with BrC who provided written informed consent, 275 did not return QLQ-C30 and BR45 questionnaires after two reminders. Amongst the 428 patients who returned their questionnaires, seven were found to be ineligible. Thirteen patients who answered fewer than 50% of the questionnaire items were excluded. Thus, 408 patients were included in the study (S1 Fig).

Patients were excluded if they had non-invasive BrC, anther primary cancer within the prior five years, or did not understand English and did not have a translator.

Patients were stratified into one of five *a priori* mutually-exclusive health states, to ensure that our sample included patients from the spectrum of BrC [19, 20]:

I: first year after diagnosis of primary BrC;

R: first year after date of local recurrence, or new primary BrC;

II-V: second to fifth year after primary BrC or local recurrence treated with curative intent;

VI+: sixth and following years after a primary BrC or local recurrence treated with curative intent;

M: metastatic BrC.

Lidgren found that EQ-5D and TTO utility instruments differentiated between most health states except for between first year of local recurrence and second year and following years after primary or local recurrent BrC [19]. We adopted Torres et al.'s (unpublished) VI+ health state to account for recent guidelines recommending adjuvant endocrine therapy for up to 10 years for women with hormone receptor positive BrC [21].

A subset of patients with BrC (n = 81) rated the importance of all items in QLQ-C30 and BR45 on a five-point scale (0—not applicable, 5 –very important).

**Clinicians.** Thirteen clinicians working with women with BrC rated the importance of QLC-C30 and BR45 as applicable to patients on a five-point scale (0—not applicable, 5 –very important) using a secure web-form. Demographic characteristics were not collected from clinicians to protect their anonymity.

## Ethics

This study was approved by three research ethics boards (ID): Sunnybrook Health Sciences Centre (1796), University Health Network (18–5350), and the University of Toronto (36324). All participants provided written informed consent.

## Statistical analysis—confirmatory factor analysis

Fig 1 shows our CFA process. We considered clinical relevance, item distributions, missing data, item-importance, and internal reliability of items within subscales of the QLQ-C30 and BR45 to create our *a priori* dimensions of our CFA. We started with King et al's process of prioritizing dimensions in the QLU-C10D [9] with World Health Organization's (WHO)'s core health dimensions [22] and cancer-specific dimensions. The WHO dimensions specific to the QLQ-C30 functioning dimensions were: physical, emotional, and social [22]. General cancer and BrC-specific dimensions were agreed by our multidisciplinary research team with expertise in patient outcome measurement, biostatistics, health economics, general internal medicine, and breast medical oncology. General cancer dimensions were: pain, fatigue (energy). BrC-dimensions were: breast and arm symptoms, sexuality, systemic therapy, endocrine therapy, and body image. The set of attributes used to develop a preference-based instrument should be both comprehensive (contain sufficient and clinically-relevant factors) and parsimonious (have a limited number of factors to minimize cognitive burden [23] for those estimating the multi-attribute utility function of the future BUI).

Since there were two sex-related dimensions, patient-rated item-importance ratings were ranked and prioritized over clinician-rated item-importance ratings to select the most important dimension to patients.

To construct the measurement model which consists of observed variables (items) and unobserved variables (dimensions), at least two items are needed to estimate each dimension, therefore, only multiple-item dimensions were included in the CFA [24]. We also removed global HRQoL items, because they are not related to particular dimensions of HRQoL [25, 26].

We conducted preliminary analyses prior to the CFA. First, the item response distributions of the QLQ-C30 and BR45 over BrC health states were visualized using stacked bar plots (S2 and S3 Figs). Next, correlations were inspected: inter-item, item-to-dimension (subscale), and inter-dimensional (S4 Fig), to ensure there was a sufficient association between items and

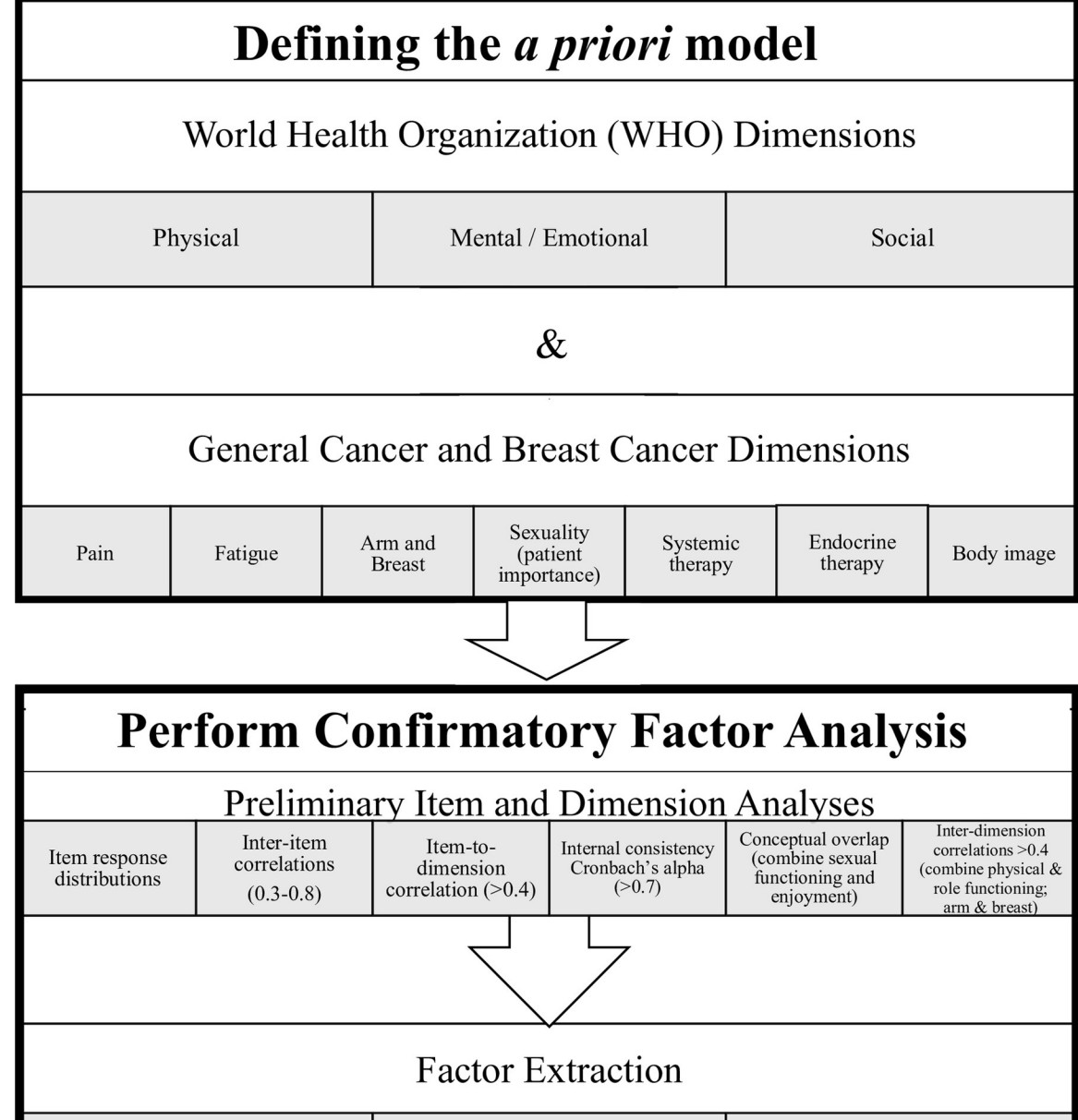

**Fig 1. Confirmatory factor analysis process.**

dimensions to move forward with CFA. Polychoric correlations were used for ordinal response options of the items [27]. We assessed correlations using the following criteria:

1. Inter-item correlations of 0.3–0.8, indicating high correlations [18],

2. Item-to-dimension correlations of >0.4, suggesting convergent validity of items within the same subscale [18],

3. Inter-dimension correlations of >0.4, supporting convergent validity.

Internal consistency, a measure of reliability, was evaluated using Cronbach's alpha for items within each hypothesized dimension based on the QLQ-C30 and BR45 scoring manuals. A target value of Cronbach's alpha >0.7 for group comparisons represented high internal consistency reliability [18, 28].

Preliminary item and dimensional analyses led to *a priori* dimension combinations or reallocation of an item (S4 Fig). We combined physical and role functioning and arm and breast symptoms based on conceptual overlap and high inter-dimension correlation (0.67 and 0.58, respectively) suggesting convergent validity. The sexual functioning and sexual enjoyment dimensions overlapped in content, and its items all had high inter-item correlations > 0.60, therefore we decided *a priori* to combine the sexual functioning and sexual enjoyment dimensions. Item 10 (needing to rest) overlapped in content with item 4 (needing to stay in bed or a chair), and both items were highly correlated (0.56), therefore we decided *a priori* to move item 10 from the Fatigue dimension to the Physical and Role functioning dimension. These combinations and item reallocations were validated by our clinician expert (MT).

Costa et al.'s factor analysis of the QLQ-C30 [25], also combined physical and role functioning due to high inter-dimensional correlation, and moved item 10 to the Physical and Role functioning dimension for the same reasons.

We performed CFA on an *a priori* 10-dimensional model to ensure a parsimonious set of attributes [23]. We used mean-and variance-adjusted unweighted least squares estimation (ULSMV) for ordinal response variables to obtain more robust model fit and standard errors and higher power, given our sample size [27, 29]. We fitted the CFA models with polychoric correlations because our item-level distributions departed from the normality assumption, and had fewer than five ordinally-scaled response variables [27, 30].

In the baseline factor model, items were specified to load onto one factor, the factors were allowed to correlate freely, and the residuals were uncorrelated, and a standardized solution was obtained.

We evaluated global model fit, saliency of parameters, and local areas of strain [31]. Nested models were compared using $\chi^2$ likelihood ratio tests. We also evaluated $R^2$, the proportion of variance in the item response explained by the factor (>0.1) [31].

Global model fit is a descriptive indicator of how well the model reproduces the observed relationships between the indicators, represented by items, in the input matrix [31]. We used five tests to evaluate global model fit:

- Sartorra Bentler (SB) scaled $\chi^2$ statistics were used if high kurtosis statistics suggested the items were not normally-distributed [32]. A non-significant SB$\chi^2$, where p> 0.05, is desired. Because the $\chi^2$ statistic is sensitive to sample size, a parsimony adjusted test statistic SB$\chi^2$/df <2 indicates a good fit [33].

- Root mean square error of approximation (RMSEA) was used to estimate the discrepancy per degree of freedom between the model implied covariance matrix and the population covariance matrix [27]. The RMSEA includes an adjustment whereby more complex models

with greater degrees of freedom are penalized. RMSEA cut-off values for fit are <0.08 (adequate), and <0.05 (good), with an upper limit of the 90% confidence interval <0.08 [34, 35].

- Standardized root mean square residual (SRMR) is the mean absolute residual correlation, where <0.08 indicates acceptable fit, and <0.06 indicates good fit [34, 35].

- Comparative fit index (CFI) and Tucker-Lewis Index (TLI) indicate improvement in fit comparing the researcher's model to the baseline exact fit ($\chi^2$) model, where >0.90 is acceptable, and >0.95 is good [34, 35].

Saliency evaluates if the items are associated with the pre-specified factor. We inspected factor loadings ($\lambda$) and considered items with $\lambda$>0.3 with statistical significance at $\alpha$ = 0.01 (with Bonferroni correction) to be salient to a given factor [36].

Where there was poor global model fit, residuals and modification indices were inspected to identify local areas of strain. Residuals are the sources of unexplained variance in the model [37]. Correlated item residuals mean that there is a common variance that is not accounted for by the initial hypothesized factor structure [37]. This common variance can occur when item content overlaps, leading to suboptimal model fit [38, 39]. A modification index approximates the degree that a model's $\chi^2$ statistic would decrease if a given fixed parameter became freely estimated, analogous to the $\chi^2$ difference (with a single degree of freedom) of nested models [31]. Therefore, well-fitted models have small modification indices. A model with local areas of strain would have item pairs in the same dimension with high residuals (>0.4), and high modification indices (>25) [40]. If there was substantial clinical rationale and overlapping item content, we re-specified models to correlate item residuals, and re-assessed residuals and modification indices.

Likelihood ratio tests were also performed to compare nested models for relative model fit. Our clinical expert (MT) checked the content validity and clinical meaningfulness of retained dimensions.

We followed the EORTC scoring manual to calculate subscale scores with missing data [41].

We performed CFA using R v1.2.5001 (http://cran.r-project.org/) using corrplot [42], psych [43], and lavaan packages [44].

## Sample size

**Patients.** Assuming six factor loadings to each factor, 400 patients were deemed to be sufficient to provide a high level of congruence (K>0.95) between the factors from the sample solution and the population solution [45].

**Clinicians.** We aimed to recruit at least 10 clinicians from a range of professions to represent different perspectives.

## Results

### Patients

Table 1 shows the demographic and clinical characteristics of patients. Patients had a mean (SD) age of 59.1 (11.6) years. The majority (64.2%) were married or in a common-law relationship, with 80% completed at least college education. Most patients were diagnosed in pathological stage 1A (37%) or IIA (25.5%), which is comparable to the incidence of BrC stages in Ontario, Canada [46, 47]. The most common treatment intents were adjuvant (64.2%), palliative (22.5%), and neoadjuvant (6.4%). The most common treatment regimens were endocrine therapy (57.0%), chemotherapy (17.1%), and targeted therapy (16.7%). Most patients were in

**Table 1. Participant characteristics and comparator population level characteristics.**

| | All patients, (N = 408) | Sub-set of all patients, item importance, (n = 81) | Population comparators⁹ | Population, reference |
|---|---|---|---|---|
| **Age** | **Years** | **Years** | **Years** | Women with breast cancer, (Ontario Canada) Seung et al. [46] |
| Mean (SD) | 59.1 (11.6) | 60.12 (11.1) | 61.5 (13.8) | |
| Range | 25–93 | | | |
| **Marital status** | **n (%)** | **n (%)** | **n (%) N = 5,812,755** | Canadian census 2016, female >15y (Ontario, Canada) [48] |
| Single | 50 (12.3) | 12 (14.8) | 1,493,605⁹⁹ (25.7) | |
| Married or common-law | 262 (64.2) | 48 (59.3) | 3,218,800 (55.4) | |
| Divorced | 36 (8.8) | 7 (8.6) | 400,935 (6.9) | |
| Separated | 16 (3.9) | 2 (2.5) | 190,535 (3.3) | |
| Widowed | 31 (7.6) | 8 (9.9) | 508,880 (8.8) | |
| Missing/did not answer | 13 (3.2) | 4 (4.9) | 0 (0) | |
| **Highest level of education** | **n (%)** | **n (%)** | **n (%) N = 5,695,685** | Canadian census 2016, female >15y (Ontario, Canada) [48] |
| Elementary school | 3 (0.7) | 1 (1.2) | 973,670 (17.1) | |
| High school | 53 (13.0) | 14 (17.3) | 1,540,770 (27.1) | |
| Trade or apprentice | 8 (2.0) | 1 (1.2) | 193,120 (3.4) | |
| College or undergraduate university | 174 (42.6) | 37 (45.7) | 2,614,965 (45.9) | |
| Graduate or professional degree | 152 (37.3) | 24 (29.6) | 373,160 (6.6) | |
| Missing/did not answer | 18 (4.4) | 4 (4.9) | 0 (0) | |
| **Years since first BrC diagnosis** | **n (%)** | **n (%)** | **n (%) N = 361** | Women with breast cancer (Stockholm, Sweden), Lidgren et al. [19] |
| < 5 | 229 (56.1) | 35 (43.2) | 183 (53.0) | |
| 5 to 9 | 91 (22.3) | 28 (34.6) | 88 (25.5) | |
| 10 to 14 | 38 (9.3) | 10 (12.3) | 74 (21.4) | |
| 15 to 19 | 19 (4.7) | 3 (3.7) | | |
| 20 to 25 | 19 (4.7) | 3 (3.7) | | |
| 25+ | 4 (1.0) | 0 (0.0) | | |
| Missing* | 8 (2.0) | 2 (2.5) | | |
| **Charlson comorbidity index (CCI)** | **n (%)** | **n (%)** | **mean (SD)** | Women with breast cancer (Ontario, Canada), Seung et al. [46] |
| 0 | 366 (89.7) | 71 (87.7) | 0.64 (1.2) | |
| 1 | 25 (6.1) | 7 (8.6) | | |
| 2 | 8 (2.0) | 1 (1.2) | | |
| 3 | 2 (0.5) | 0 (0.0) | | |
| Missing | 7 (1.7) | 2 (2.5) | | |
| **Health state**** | **n (%)** | **n (%)** | **n (%) N = 361** | Women with breast cancer (Stockholm, Sweden), development of health states Lidgren et al. [19] |
| I | 81 (19.9) | 13 (16.0) | 72 (20.9) | |
| R | 9 (2.2) | 2 (2.5) | 21 (6.1) | |
| II-V | 127 (31.1) | 29 (35.8) | 185 (53.6) | |
| VI+ | 88 (21.6) | 24 (29.6) | | |
| M | 103 (25.2) | 13 (16.0) | 67 (19.4) | |
| **Menstrual status** | **n (%)** | **n (%)** | **n (%) N = 250** | Women with breast cancer (11 European countries and Brazil), development of BR45 Bjelic-Radisic et al. [14] |
| Pre-menopausal | 40 (9.8) | 9 (11.1) | 59 (23.6) | |
| Post-menopausal | 309 (75.7) | 58 (71.6) | 178‡ (71.2) | |
| I don't know | 32 (7.8) | 7 (8.6) | 11 (4.4) | |
| Missing | 27 (6.6) | 7 (8.6) | 2 (0.8) | |
| **Pathological stage at initial surgery** | **N (%)** | **n (%)** | **n (%) N = 34,340** | Women with breast cancer (Ontario, Canada), Seung et al. [46] |
| IA | 151 (37.0) | 20 (28.6) | 13,989 (40.7) | |
| IB | 4 (1.0) | | | |
| IIA | 104 (25.5) | 28 (40.0) | 12,819 (37.3) | |
| IIB | 69 (16.9) | 10 (14.3) | | |
| IIIA | 45 (11.0) | 7 (10.0) | 4,508 (13.1) | |
| IIIB | 5 (1.2) | 0 (0) | | |
| IIIC | 15 (3.7) | 4 (5.7) | | |
| IV | 12 (2.9) | 1 (1.4) | 1,673 (4.9) | |
| No surgery / unknown | 3 (0.7) | | 1,265 (3.7) | |

(*Continued*)

**Table 1.** (Continued)

| | All patients, (N = 408) | Sub-set of all patients, item importance, (n = 81) | Population comparators⁹ | Population, reference |
|---|---|---|---|---|
| **Biomarkers** | Positive n (%) | Negative n (%) | Disease sub-type n (%) N = 34,340 | Women with breast cancer (Ontario, Canada), Seung et al. [46] |
| Estrogen receptor | 329 (47.8) | 64 (48.1) | Hormone +, HER2-: 22,247 (64.8) | |
| Progesterone receptor | 294 (42.7) | 57 (42.9) | | |
| HER-2*** receptor | 66 (9.6) | 12 (9.0) | HER2+: 4,902 (14.3) | |
| | | | Triple negative: 3,277 (9.5) | |
| | | | Unknown subtype: 3,914 (11.4) | |
| **Surgery****** | n (%) | n (%) | n (%) N = 250 | Women with breast cancer, (11 European countries and Brazil) development of BR45 Bjelic-Radisic et al. [14] |
| Breast conserving surgery | 281 (53.8) | 63 (54.8) | 117‡‡ (40.5) | |
| Mastectomy | 196 (37.5) | 35 (30.4) | 96‡‡‡ (33.2) | |
| Mastectomy: prophylactic | 26 (5.0) | 7 (6.1) | | |
| Other | 15 (2.9) | 10 (8.7) | 37 (12.8) | |
| Missing | 4 (0.8) | | 39 (13.5) | |
| **Surgery–axillary** | n (%) | n (%) | n (%) N = 250 | Women with breast cancer, (11 European countries and Brazil) development of BR45 Bjelic-Radisic et al. [14] |
| Axillary lymph node dissection (ALND) | 125 (30.6) | 23 (28.0) | 92 (50.3) | |
| Sentinel lymph node biopsy (SLNB) | 240 (58.8) | 54 (65.9) | 91 (49.7) | |
| SLNB and ALND | 11 (2.7) | 5 (6.1) | | |
| **Intent of systemic therapy** | n (%) | n (%) | | |
| Neoadjuvant | 26 (6.4) | 4 (4.9) | | |
| Adjuvant | 262 (64.2) | 58 (71.6) | | |
| Palliative | 92 (22.5) | 12 (14.8) | | |
| No treatment–active surveillance | 28 (6.9) | 7 (8.6) | | |
| **Regimen** | n (%) | n (%) | | |
| Chemotherapy | 96 (17.1) | 11 (10.6) | | |
| Endocrine therapy | 321 (57.0) | 66 (63.5) | | |
| No treatment–active surveillance | 28 (5.0) | 6 (5.8) | | |
| Radiotherapy | 17 (3.0) | 4 (3.8) | | |
| Targeted therapy | 94 (16.7) | 14 (13.5) | | |
| Zoledronic acid only | 7 (1.2) | 3 (2.9) | | |
| **Chemotherapy** | n (%) | | | |
| Abraxane weekly | 2 (2.1) | | | |
| Doxorubicin (Adriamycin and Cyclophosphamide, AC) | 1 (1.0) | | | |
| Doxorubicin low dose | 5 (5.2) | | | |
| AC>Paclitaxel dose dense | 16 (16.5) | | | |
| Capecitabine | 19 (19.6) | | | |
| Cisplatin and Gemcitabine | 1 (1.0) | | | |
| Cyclophosphamide and methotrexate oral | 1 (1.0) | | | |
| Docetaxel and cyclophosphamide | 4 (4.1) | | | |
| Docetaxel weekly | 1 (1.0) | | | |
| Peglyated liposomal doxorubicin | 1 (1.0) | | | |
| Eribulin | 2 (2.1) | | | |
| Etoposide | 1 (1.0) | | | |
| Fluorouracil epirubicin cyclophosphamide (FEC100) >Docetacel (Doc100) | 25 (25.8) | | | |
| Paclitaxel | 14 (14.4) | | | |
| Docetaxel (Taxotere) and Cyclophosphamide | 4 (4.1) | | | |
| **Targeted therapy** | n (%) | | | |
| Palbociclib | 36 (34.6) | | | |
| Ribociclib | 4 (3.8) | | | |

*(Continued)*

**Table 1.** (Continued)

| | All patients, (N = 408) | Sub-set of all patients, item importance, (n = 81) | Population comparators[¶] | Population, reference |
|---|---|---|---|---|
| Pertuzumab | 18 (17.3) | | | |
| Trastuzumab | 38 (36.5) | | | |
| Capecitabine and Lapatinib | 1 (1.0) | | | |
| Capecitabine and Trastuzumab | 1 (1.0) | | | |
| Lapatinib | 1 (1.0) | | | |
| Kadcyla | 2 (1.9) | | | |
| Everolimus and Exemestane | 1 (1.0) | | | |
| Venetoclax | 2 (1.9) | | | |
| **Endocrine therapy** | **n (%)** | | | |
| Letrozole | 96 (43.4) | | | |
| Tamoxifen | 85 (38.5) | | | |
| Anastrazole | 69 (31.2) | | | |
| Exemestane | 34 (15.4) | | | |
| Fulvestrant | 8 (3.6) | | | |
| Leuprolide | 7 (3.2) | | | |
| Goserelin | 18 (8.1) | | | |
| **Other–bone modifying agents** | **n (%)** | | | |
| Denosumab | 16 (14.8) | | | |
| Zoledronic acid | 92 (85.2) | | | |
| **Radiotherapy intent** | **n (%)** | | | |
| Adjuvant | 15 (83.3) | | | |
| Palliative | 3 (16.7) | | | |

[¶]Population comparators were mostly from women with breast cancer, except marital status and highest level of education comparators were drawn from the 2016 Canadian Census.

[¶¶]2016 Canadian census data: never married.

[‡]150 post-menopausal and 28 treatment-related menopause, total of 178.

[‡‡]104 breast conserving surgeries and 13 oncoplastic breast conserving surgeries, total of 117.

[‡‡‡]49 simple mastectomies and 47 mastectomies and reconstruction surgeries, total of 96.

[*]Referral from another centre. Date and month were approximate.

[**] Mutually-exclusive health states: 1) first year after primary BrC diagnosis treated with curative intent (I), second to fifth year after primary BrC diagnosis (II-V), sixth year onwards (VI), metastatic diseases (M), local recurrence of BrC (R).

[***] 5 breast tumours were HER-2 equivocal.

[****]405 patients had a combined 522 surgeries. Three patients did not receive surgery.

their second to fifth year health state (31.1%). A higher proportion of patients in our sample had metastatic disease than a development study of the BrC health states (25.2% vs 19.4%) [19].

The subset of patients (n = 81) who rated item-importance were of comparable age, bio-marker status, comorbidity status as all participants (Table 1). The item-importance sample consisted of a smaller percentage than the full sample with a graduate or professional degree (29.6% vs 37.3%), fewer in the metatstatic health state (16.0% vs 25.2%); and, a larger percentage were diagnosed with BrC from 5 to 9 years (34.6% vs 22.3%).

The 13 clinicians who completed importance ratings were five medical oncologists, one radiation oncologist, one surgical oncologist, two medical oncology fellows, two nurses, one

**Table 2. QLQ-C30 and BR45 subscale and internal consistency scores.**

| Instrument subscale | Subscale scores (0 to 100) Higher scores for symptoms imply more severe symptoms, while higher scores for functioning imply greater ability. | | Internal consistency |
|---|---|---|---|
| | mean +/- SD | median (1st quartile, 3rd quartile) | Cronbach's alpha |
| QLQ-C30 | | | |
| Physical functioning (PF) | 83.9 +/- 16.93 | 86.67 (73.33, 100) | 0.79* |
| Role functioning (RF) | 79.72 +/-24.69 | 83.33 (66.67, 100) | 0.85* |
| Emotional functioning (EF) | 73.45 +/-21.70 | 75.00 (58.33, 91.67) | 0.86 |
| Cognitive functioning (CF) | 79.16 +/-21.48 | 83.33 (66.67, 100) | 0.70 |
| Social functioning (SF) | 77.13 +/- 27.15 | 83.33 (66.67, 100) | 0.90 |
| Fatigue (FA) | 31.51 +/- 22.76 | 33.33 (11.11, 44.44) | 0.86 |
| Nausea, vomiting (NV) | 4.45 +/- 10.74 | 0 (0, 0) | 0.41 |
| Pain (PA) | 23.67 +/- 25.95 | 0 (0, 33.33) | 0.85 |
| Single item subscales | | | |
| Dyspnea (DY) | 12.30 +/- 20.40 | 0 (0, 33.33) | |
| Insomnia (SL) | 33.09 +/- 29.73 | 33.33(0, 66.67) | |
| Appetite loss (AP) | 11.49 +/- 21.12 | 0 (0, 33.33) | |
| Constipation (CO) | 15.37 +/- 24.41 | 0 (0, 33.33) | |
| Diarrhea (DI) | 9.30 +/- 19.38 | 0 (0, 0) | |
| BR45 | | | |
| Body image (BI) | 67.93 +/- 29.67 | 75 (50, 91.67) | 0.91 |
| Future perspective (FU) | 44.66 +/- 31.82 | 33.33 (33.33, 66.67) | |
| Sexual functioning (SX) | 82.12 +/- 21.41 | 83.33 (66.67, 100) | 0.76** |
| Sexual enjoyment (SE) | 73.00 +/- 31.30 | 100 (66.67, 100) | |
| Breast satisfaction (BS) | 43.09 +/- 32.40 | 33.33 (16.67, 66.67) | 0.91 |
| Systemic therapy side effects (SYS) | 21.86 +/- 16.89 | 19.05 (9.52, 33.33) | 0.69 |
| Upset by hair loss (HU) | 44.07 +/- 36.44 | 33.33 (0, 66.67) | |
| Arm symptoms (ARM) | 20.58 +/- 22.64 | 11.11 (0, 33.33) | 0.77*** |
| Breast symptoms (BR) | 17.01 +/- 18.68 | 16.67 (0, 25) | 0.78 |
| Endocrine therapy symptoms (ET) | 25.64 +/- 19.12 | 23.33 (10.00, 36.67) | 0.85 |
| Skin mucositis symptoms (SM) | 13.50 +/- 14.79 | 11.11 (0, 22.22) | 0.74 |
| Endocrine sexual symptoms (ES) | 28.24 +/- 30.06 | 16.67 (0, 50.00) | 0.94 |

**Cronbach's alpha of combined subscales:** *PF + RF + item 10 = 0.86; **SX+SE = 0.83; ***ARM+BR = 0.83

physician assistant, and one social worker, predominantly representing the medical oncology clinical staff.

The item response distributions by subscale are shown in S2 and S3 Figs. Table 2 describes subscale scores on the QLQ-C30 and BR45. Our patients' QLQ-C30 subscale scores were between the EORTC reference values of patients with early stage and metastatic BrC [49]. Reference values for BR45 scores are not yet available. Cronbach's alpha for most subscales were greater than our cut-off of 0.70 (Table 2). Based on consultation with our clinical expert (MT), the factors with the lowest alpha were removed (e.g., nausea and vomiting, $\alpha = 0.41$), or kept in the CFA model because of clinical significance (e.g., systemic therapy side effects, $\alpha = 0.69$).

## Missing responses, and removal of dimensions or items

Only raw scores were used in the CFA. If a patient completed at least 50% of the items in the dimension, the missing item(s) were imputed as the mean of the scale items the patient answered to calculate the QLQ-C30 and BR45 scores (Table 2) [41]. Most items had less than 2% missing, except for the sex-related items, which 17–51% of patients omitted (S1 Table).

## Dimension and item importance rated by patients and clinicians

The mean importance ratings by patients and clinicians are shown in the Table 3. In 15/26 dimensions, clinician ratings were significantly higher than patients (p<0.05), otherwise, ratings were similar between the two groups. The sexual functioning dimension was rated significantly higher by patients than clinicians (4.22 vs 3.59, p 0.002). The three sex-related dimensions had mean patient-rated dimensional ratings of 4.22, 4.03, 3.94, for sexual functioning, sexual enjoyment, and endocrine sexual symptoms, respectively. These patient-rated importance ratings supported our *a priori* retention of the combined sexual functioning and sexual enjoyment dimension.

Table 3. Mean dimension importance rated by patients and clinicians.

| QLQ-C30 Dimension | Mean patient importance of dimension[‡] (n = 81) | Mean clinician importance of dimension[‡] (n = 13) | Welch two sample t-test p-value |
|---|---|---|---|
| Physical functioning | 3.73 | 4.00 | 0.047 |
| Role functioning | 3.44 | 4.00 | 0.002 |
| Emotional functioning | 3.41 | 4.10 | <0.001 |
| Cognitive functioning | 3.39 | 4.04 | <0.001 |
| Social functioning | 3.61 | 3.96 | 0.114 |
| Fatigue | 3.29 | 3.90 | <0.001 |
| Nausea and vomiting | 3.50 | 4.13 | 0.418 |
| Pain | 3.52 | 4.27 | <0.001 |
| Dyspnea | 3.33 | 4.00 | 0.037 |
| Insomnia | 3.54 | 3.83 | 0.263 |
| Appetite loss | 3.67 | 3.75 | 0.845 |
| Constipation | 3.86 | 3.50 | 0.322 |
| Diarrhea | 4.50 | 3.67 | 0.312 |
| Financial difficulties | 3.27 | 4.07 | 0.004 |
| **BR45 Dimension** | | | |
| Body image | 3.51 | 3.74 | 0.109 |
| Future perspective | 3.62 | 4.08 | 0.085 |
| Sexual functioning | 4.22 | 3.59 | 0.002 |
| Sexual enjoyment | 4.03 | 3.73 | 0.236 |
| Breast satisfaction | 3.42 | 3.86 | 0.032 |
| Systemic therapy side effects | 3.64 | 3.75 | 0.31 |
| Upset by hair loss | 3.81 | 4.42 | 0.018 |
| Arm symptoms | 3.26 | 3.82 | <0.001 |
| Breast symptoms | 3.33 | 3.77 | 0.004 |
| Endocrine therapy symptoms | 3.70 | 3.95 | 0.004 |
| Skin mucositis symptoms | 3.37 | 3.81 | <0.001 |
| Endocrine sexual symptoms | 3.94 | 3.84 | 0.466 |

[‡]Importance scores were: 0—not applicable, 1—slight, 2—mild, 3—moderate, 4—important, 5—very important. Mean importance = (sum importance / number of items).

Given that 30% of items were rated scores 0 (not applicable), 1 (slight) and 2 (mild), we removed these scores prior to analysis to de-emphasize mild but frequent aspects of HRQoL [50].

## Confirmatory factor analysis

Table 4 shows the *a priori* model. It includes items from the following dimensions of the QLQ-C30: physical and role functioning, pain, fatigue, emotional functioning, social

**Table 4. *A priori* factor model–factors and item summary.**

| Physical and role functioning | Pain | Fatigue | Emotional functioning | Social functioning |
|---|---|---|---|---|
| PF1. Trouble doing strenuous activities. | During the past week: | During the past week: | During the past week: | During the past week: |
| PF2. Trouble taking a long walk. | PA9. Had pain. | FA12. Felt weak. | EF21. Felt tense. | SF26. Physical condition or medical treatment interfered with your family life. |
| PF3. Trouble taking a short walk. | PA19. Pain interfered with your daily activities. | FA18. Tired. | EF22. Worry. | SF27. Physical condition or medical treatment interfered with your social activities. |
| PF4. Need to stay in bed or a chair during the day. | | | EF23. Felt irritable. | |
| PF5. Need help with eating, dressing, washing yourself or using the toilet. | | | EF24. Felt depressed. | |
| RF6. Limited in doing either your work or other daily activities | | | | |
| RF7. Limited in pursuing your hobbies or other leisure time activities. | | | | |
| FA10. Need to rest. | | | | |
| **Systemic therapy side effects** | **Body image** | **Sexual functioning and enjoyment** | **Arm and breast symptoms** | **Endocrine therapy symptoms** |
| SYS31. Dry mouth. | BI39. Felt physically less attractive as a result of your disease or treatment. | SX44. Interested in sex. | ARM47. Pain in your arm or shoulder. | ET54. Sweated excessively. |
| SYS32. Food and drink tasted different than usual. | BI40. Felt less feminine as a result of your disease or treatment. | SX45. Sexually active (with or without intercourse). | ARM48. Swollen arm or hand. | ET55. Had mood swings. |
| SYS33. Eyes been painful, irritated or watery. | BI41. Problems looking at yourself naked. | SE46. Sex been enjoyable. | ARM49. Problems raising your arm or moving it sideways. | ET56. Dizzy. |
| SYS34. Lost any hair | BI42. Dissatisfied with your body. | | BR50. Pain in the area of your affected breast. | ET63. Problems with your joints. |
| SYS36. Felt ill or unwell | | | BR51. Area of your affected breast been swollen. | ET64. Stiffness in your joints. |
| SYS37. Hot flushes | | | BR52. Area of your affected breast been oversensitive. | ET65. Pain in your joints. |
| SYS38. Had headaches | | | BR53. Skin problems on or in the area of your affected breast (e.g., itchy, dry, flaky). | ET66. Aches or pains in your bones. |
| | | | | ET67. Aches or pains in your muscles. |
| | | | | ET68. Gained weight |
| | | | | ET69. Weight gain been a problem for you. |

**Table 5. Summary of robust fit statistics of CFA models.**

| | $\chi^2$/df[+] | TLI[++] | CFI[++] | SRMR[+++] | RMSEA[++++] (90% CI) | $\chi^2$ test of nested models |
|---|---|---|---|---|---|---|
| **Model A**<br>All items in original dimension. | 1.70 | 0.917 | 0.923 | 0.080 | 0.042 (0.038–0.045) | Model B compared with Model A<br>$\Delta$SB$\chi^2$ = 573.4<br>p<0.001 |
| **Model B**<br>3 items moved to different dimensions*. | 1.53 | 0.937 | 0.942 | 0.073 | 0.036 (0.033–0.039) | |
| **Model C**<br>3 items moved to different dimensions *.<br>3 residual correlations**. | 1.45 | 0.946 | 0.951 | 0.069 | 0.033 (0.030–0.037) | Model C compared with Model B<br>$\Delta$SB$\chi^2$ = 225.62<br>p < 0.001 |

+ $\chi^2$/df <2 indicate good model fit.

++Tucker-Lewis Index (TLI) and Comparative fit index (CFI), >0.95 is good.

+++Standardized root mean square residual (SRMR) <0.08 acceptable.

++++Root mean square error of approximation (RMSEA) < 0.08 adequate, with 90% CI of upper CI <0.07.

*3 items moved to different dimensions were

ET55 (mood swings), moved from Endocrine Therapy to Emotional Functioning, SYS37 (hot flushes) moved from Systemic Therapy to Endocrine Therapy, ET56 (dizziness) moved from Endocrine Therapy to Fatigue

**3 residual correlations were applied between items PF2 (long walk) and PF3 (short walk); ET68 (gained weight) and ET69 (weight gain has been a problem); SYS37 (hot flushes) and ET54 (sweated excessively).

functioning; and, from dimensions of the BR45: systemic therapy side effects, body image, sexual functioning and enjoyment, arm and breast symptoms, and endocrine therapy symptoms.

The results from our CFA are presented in Table 5, showing a summary of robust fit indices after re-specifying the original model and after applying residual correlations. The baseline model had inadequate global model fit (Model A), and local areas of strain represented by highly correlated residuals (>0.3) and high modification indices (> 25). After re-specifications of the model, we obtained adequate global model fit (Model C). Modifications to the baseline model involved moving three items to different dimensions (Model B), and applying three pairs of residual correlations to reduce local areas of strain (Model C).

The modification indices supported high error covariances between item pairs and between several items and specific dimensions. The three largest and most significant modification indices (MI) between item pairs (> 25) [40] were consistently present in tested models. These same item pairs also exhibited high residual correlations (>0.3): PF2 (long walk) and PF3 (short walk); ET68 (gained weight) and ET69 (weight gain has been a problem); SYS37 (hot flushes) and ET54 (sweated excessively). These three item pairs with correlated residuals involved similar functional limitations and were within the same dimension. To reduce the high MIs between items and dimensions, SYS37 (hot flushes) was moved from the Systemic Therapy Side Effects dimension to the Endocrine Therapy dimension; ET55 (mood swings) was moved from the Endocrine Therapy Symptoms dimension to the Emotional Functioning dimension; and ET56 (dizziness) was moved from the Endocrine Therapy Symptoms dimension to the Fatigue dimension. These item re-assignments were approved by our clinical expert (MT). After re-assigning these items to the aforementioned dimensions, there were no high MIs.

Nested models were compared in likelihood ratio tests (model B vs A; model C vs B) (Table 5). The fit indices of the refined model with three correlated item residuals demonstrated significant improvements in fit compared with the *a priori* model. The re-specified model with SYS37 (hot flushes), ET55 (mood swings), and ET56 (dizziness) reassigned to Endocrine Therapy, Emotional Functioning and Fatigue dimensions, respectively (model B) demonstrated improved fit over the model with all items in their original dimensions (model

i)  QLQ C30 dimensions

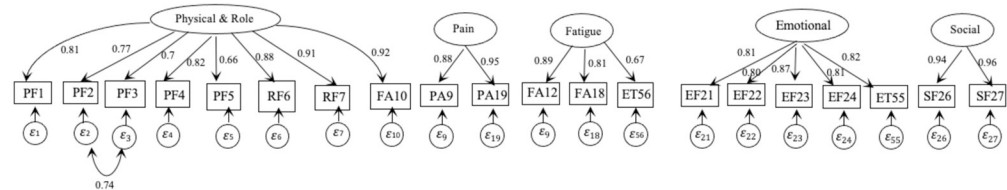

ii)  BR45 dimensions

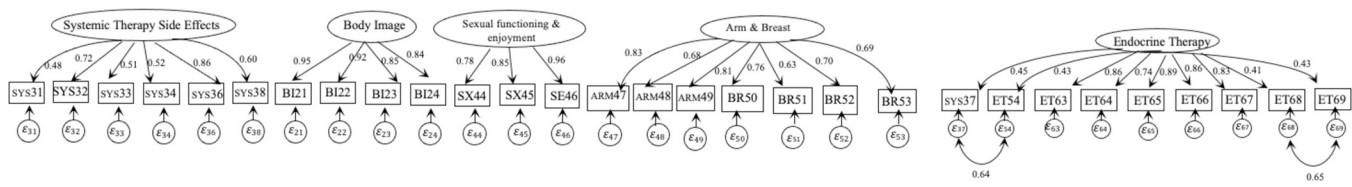

**Fig 2. Diagram of final ten-dimension CFA model (model C).**

A). Furthermore, model C performed significantly better than model B ($\Delta SB\chi^2$ = 225.62, p <0.001).

Considering our goal of identifying the most parsimonious and best-fitting model, we chose model C as our final model. Fig 2 shows the final CFA of model C with standardized factor loadings. The factor loadings and proportions of variance explained by each item are presented in Table 6. This shows that all items have a factor loading >0.4 (all p<0.001). Items had an $R^2$ ranging from 0.167 to 0.923, which are acceptable (>0.1).

**Table 6. Final model factor loadings and proportion of variance of the item responses explained by the specific factor.**

| Dimension and item topic | Model C | |
|---|---|---|
| | Factor loadings* | Proportion of variance explained |
| **Physical and role functioning** | | |
| PF1. Trouble doing strenuous activities. | 0.807 | 0.651 |
| PF2. Trouble taking a long walk. | 0.773 | 0.597 |
| PF3. Trouble taking a short walk. | 0.704 | 0.496 |
| PF4. Need to stay in bed or a chair during the day. | 0.817 | 0.668 |
| PF5. Need help with eating, dressing, washing yourself or using the toilet. | 0.655 | 0.429 |
| During the past week: | | |
| RF6. Limited in doing either your work or other daily activities. | 0.884 | 0.782 |
| RF7. Limited in pursuing your hobbies or other leisure time activities. | 0.908 | 0.824 |
| FA10. Need to rest. | 0.919 | 0.845 |
| **Pain** | | |
| During the past week: | | |
| PA9. Had pain. | 0.877 | 0.770 |
| PA19. Pain interfered with your daily activities. | 0.955 | 0.912 |

(*Continued*)

**Table 6.** (Continued)

| Dimension and item topic | Model C | |
|---|---|---|
| | Factor loadings* | Proportion of variance explained |
| **Fatigue** | | |
| During the past week: | | |
| FA12. Felt weak. | 0.886 | 0.784 |
| FA18. Tired. | 0.615 | 0.664 |
| ET 56. Dizzy. | 0.667 | 0.445 |
| **Emotional functioning** | | |
| During the past week: | | |
| EF21. Felt tense. | 0.810 | 0.656 |
| EF22. Worry. | 0.801 | 0.642 |
| EF23. Feel irritable. | 0.870 | 0.756 |
| EF24. Feel depressed. | 0.814 | 0.662 |
| ET55. Had mood swings. | 0.822 | 0.675 |
| **Social functioning** | | |
| During the past week: | | |
| SF26. Physical condition or medical treatment interfered with your family life. | 0.937 | 0.877 |
| SF27. Physical condition or medical treatment interfered with your social activities. | 0.961 | 0.923 |
| **Systemic therapy side effects** | | |
| During the past week: | | |
| SYS31. Dry mouth. | 0.481 | 0.231 |
| SYS32. Food and drink tasted different than usual. | 0.722 | 0.521 |
| SYS33. Eyes been painful, irritated or watery. | 0.512 | 0.262 |
| SYS34. Lost any hair. | 0.525 | 0.275 |
| SYS36. Felt ill or unwell. | 0.860 | 0.740 |
| SYS38. Had headaches. | 0.596 | 0.356 |
| **Body image** | | |
| BI39. Felt physically less attractive as a result of your disease or treatment. | 0.953 | 0.909 |
| BI40. Felt less feminine as a result of your disease or treatment. | 0.919 | 0.844 |
| BI41. Problems looking at yourself naked. | 0.848 | 0.719 |
| BI42. Dissatisfied with your body. | 0.840 | 0.706 |
| **Sexual functioning and enjoyment** | | |
| During the past four weeks: | | |
| SX44. Interested in sex. | 0.785 | 0.615 |
| SX45. Sexually active (with or without intercourse). | 0.845 | 0.715 |
| SE46. Sex been enjoyable. | 0.958 | 0.917 |
| **Arm and breast symptoms** | | |
| During the past week: | | |
| ARM47. Pain in your arm or shoulder. | 0.825 | 0.681 |
| ARM48. Swollen arm or hand. | 0.684 | 0.467 |
| ARM49. Problems raising your arm or moving it sideways. | 0.810 | 0.656 |
| BR50. Pain in the area of your affected breast. | 0.756 | 0.572 |
| BR51. Area of your affected breast been swollen. | 0.634 | 0.402 |
| BR52. Area of your affected breast been oversensitive. | 0.699 | 0.489 |

(*Continued*)

**Table 6.** (Continued)

| Dimension and item topic | Model C | |
|---|---|---|
| | Factor loadings* | Proportion of variance explained |
| BR53. Skin problems on or in the area of your affected breast (e.g., itchy, dry, flaky). | 0.695 | 0.483 |
| **Endocrine therapy symptoms** | | |
| SYS37. Hot flushes. | 0.454 | 0.206 |
| ET54. Sweated excessively. | 0.427 | 0.182 |
| ET63. Problems with your joints. | 0.864 | 0.747 |
| ET64. Stiffness in your joints. | 0.745 | 0.555 |
| ET65. Pain in your joints. | 0.887 | 0.787 |
| ET66. Aches or pains in your bones. | 0.864 | 0.747 |
| ET67. Aches or pains in your muscles. | 0.826 | 0.683 |
| ET68. Gained weight. | 0.409 | 0.167 |
| ET69. Weight gain been a problem for you. | 0.427 | 0.183 |

* All factor loadings were significant at p<0.001

For ease of visualization, the QLQ-C30 and BR45 dimensions are shown separately even though all ten dimensions were fitted in one CFA model. Ovals represent factors; boxes represent items; arrows with numbers represent factor loadings between factors, circles represent residuals (error term).

Residual correlations are shown within the Physical & Role Functioning dimension (between PF2 and PF3) and Endocrine Therapy dimension (between SYS37 and ET54, and ET68 and ET69).

## Discussion

This is the first CFA model of the EORTC QLQ-C30 and its novel BrC module, BR45, also the first key step of developing the Breast Utility Instrument (BUI). Our contributions to the literature lie in our methodological approach to developing a novel BrC-specific preference-based instrument, the BUI. While a CFA begins with a strong hypothesis about the dimensional structure, methodologists still recommend testing competing models [32]. Similarly, since our *a priori* model did not have good model fit, after minor modification to the specification of three items to different dimensions, and applying three residual correlations, we were able to define a model with acceptable fit of BrC-related HRQoL.

The final model fit the *a priori* dimensions including the WHO's conceptualization of health, and key BrC-specific dimensions from the EORTC QLQ-C30 and BR45 [14]. A similar approach was used to derive the general cancer utility instruments QLU-C10D from the QLQ-C30 [51] and the FACT-8D from the FACT-G [52].

All authors who performed CFA of the QLQ-C30, alone or with the previous BrC module, BR23, tested hypotheses based on the scoring manual [53], consulted patients' and clinicians' perspectives from literature searches [25, 54], or determined the core dimensions based on investigator consensus [51]. Other authors took an EFA approach, without starting from an *a priori* theoretical framework which specifies item alignment with latent variables [25, 26].

Despite not starting from the same *a priori* theoretical framework, our CFA model of the QLQ-C30 and BR45 is closely aligned with previous factor models of the BR23, while

including updated treatment-related symptoms in the BR45: systemic therapy side effects, sexual functioning and enjoyment, and endocrine therapy symptoms.

Our patient characteristics suggest that the BUI may be more applicable to long-term survivors of BrC on adjuvant endocrine therapy or patients who have metastatic disease, than those with early-stage BrC or undergoing chemotherapy. Most of our patients were on adjuvant systemic therapy (64.2%), or endocrine therapy (57%), and 24% were on any chemotherapy (Table 1). In comparison, 71.9% of patients in the development study for BR45 were on taxane chemotherapy, and 64.7% and 62.1% were taking cyclophosphamide or anthracycline, respectively [14].

We balanced comprehensive coverage of relevant factors and items with adequate model fit, sacrificing comprehensiveness to achieve parsimony and global model fit. We removed single-item functioning (global QoL, future perspective) and symptom subscales (e.g., dyspnea, insomnia, upset by hair loss). When dimensions overlapped, we chose the dimension rated more important by patients, i.e., retained sexual functioning and enjoyment. We excluded the nausea and vomiting dimension because of low internal consistency ($\alpha = 0.4$) and low patient-rated importance (3.50). Few patients reported nausea and vomiting, likely because 17.1% of patients were on chemotherapy.

We prioritized patient experience with the sex-related dimension *a priori* because patients have first-hand experience of the illness, are well informed about the burden of disease, and have experience undergoing treatment [55]. Patients may rate the importance of dimensions as lower than clinicians, because patients are known to adapt to their health conditions, including changing their internal standards and values [56, 57].

Our re-specified models considered clinical relevance when reducing local areas of strain. ET55 (mood swings), originally in the Endocrine Therapy dimension, had a high modification index with the Emotional Functioning dimension, so re-specifying the model with ET55 in the Emotional Functioning dimension was congruent. SYS37 (hot flushes) and ET54 (sweating excessively) had high item residuals and a high modification index. These climacteric symptoms could be more predominant in tamoxifen and ovarian function suppression [58], so were therefore moved to the Endocrine Therapy dimension. ET56 (dizziness), originally in the endocrine therapy dimension had a high modification index with the fatigue dimension. While the diagnoses of dizziness can generally be one of four types: vertigo, disequilibrium, pre-syncope, or light-headedness [59], a high MI between dizziness and the fatigue dimension suggested that our participants more often associated dizziness with fatigue.

All of the patients and clinicians in our study were accrued from one urban cancer centre. To mitigate this lack of generalizability in the development sample, future validation of the dimensional structure will ideally include responses from patients and clinicians from multiple hospital sites. Patients should represent a wider range of treatments spanning all five health states. Other developers of condition-specific preference-based instruments involved patients and clinicians to validate their *a priori* dimensions, namely, the QLU-C10D for general cancer [9], DUI for diabetes [60], and NQU for multiple sclerosis [61].

## Conclusions

The results of this CFA established the dimensional structure and is a first step to developing the BUI, a BrC preference-based instrument. The next steps in developing the BUI will focus on selecting the core dimensions (attributes) and most representative items per dimension [12].

Overall, understanding the dimensional structure of a novel psychometric questionnaire contributes to the development of a novel condition-specific preference-based instrument.

The BUI, derived from the EORTC QLQ-C30 and BR45, will incorporate patient preferences to improve clinical and policy decisions.

## Supporting information

**S1 Fig. Participant flow diagram.**
(PDF)

**S2 Fig. Item response distributions of EORTC QLQ-C30 subscales by BrC health state.**
(PDF)

**S3 Fig. Item response distributions on EORTC QLQ BR45 subscales by BrC health state.**
(PDF)

**S4 Fig. Inter-subscale correlations proportional to colour intensity and dot size.**
(PDF)

**S1 Table. QLQ-C30 and BR45 items, associated scale, percentage missing.**
(PDF)

## Acknowledgments

Medical oncology clinicians who helped with patient recruitment: Andrea Eisen, Kataryna Jerzak, Rossanna Pezo, Sonal Gandhi, Ellen Warner, Danilo Giffoni, Lisa Verity, Elizabeth Matheson, Kim Nguyen, Neda Stjepanovic, and Sunnybrook nursing staff.

Biomatrix support: Kathy Pritchard, Nim Li, Cordelia He, Martin Yaffe.

THETA members who provided input on patient recruitment: Suzanne Chung, Josephine Wong, Chang-Ho Lee.

Patient chart data abstraction: Arcturus Phoon.

## Author Contributions

**Conceptualization:** Teresa C. O. Tsui, Maureen Trudeau, Nicholas Mitsakakis, Sofia Torres, Murray D. Krahn.

**Data curation:** Teresa C. O. Tsui, Nicholas Mitsakakis, Sofia Torres, Karen E. Bremner, Doyoung Kim.

**Formal analysis:** Teresa C. O. Tsui, Aileen M. Davis.

**Funding acquisition:** Teresa C. O. Tsui, Maureen Trudeau, Murray D. Krahn.

**Investigation:** Teresa C. O. Tsui, Maureen Trudeau, Aileen M. Davis, Murray D. Krahn.

**Methodology:** Teresa C. O. Tsui, Nicholas Mitsakakis, Karen E. Bremner, Aileen M. Davis, Murray D. Krahn.

**Project administration:** Teresa C. O. Tsui, Karen E. Bremner, Doyoung Kim.

**Resources:** Maureen Trudeau, Sofia Torres, Doyoung Kim, Murray D. Krahn.

**Software:** Teresa C. O. Tsui, Nicholas Mitsakakis.

**Supervision:** Maureen Trudeau, Nicholas Mitsakakis, Aileen M. Davis, Murray D. Krahn.

**Validation:** Teresa C. O. Tsui, Maureen Trudeau, Sofia Torres.

**Visualization:** Teresa C. O. Tsui.

**Writing – original draft:** Teresa C. O. Tsui.

**Writing – review & editing:** Teresa C. O. Tsui, Maureen Trudeau, Nicholas Mitsakakis, Sofia Torres, Karen E. Bremner, Aileen M. Davis, Murray D. Krahn.

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
