## [Decision Letter · Decision Letter 0]

24 Sep 2021

PONE-D-21-27345Confirmatory factor analysis of the EORTC QLQ-C30 and BR45 to develop the Breast Utility Instrument, a preference-based instrument to measure health related quality of life in women with breast cancerPLOS ONE

Dear Dr. Tsui,

Thank you for submitting your manuscript to PLOS ONE. After careful consideration, we feel that it has merit but does not fully meet PLOS ONE’s publication criteria as it currently stands. Therefore, we invite you to submit a revised version of the manuscript that addresses the points raised during the review process.

My personal opinion is that the manuscript is not easy to follow by readers and researchers who are not strictly familiar with statistical and psychometric  issues. Therefore, I would appreciate a tentative of  explaining better some technical aspects. I am a psychometrician aware of the difficulty that many colleagues who are expert in other fields may have, and I would like to favour them appreciating your clinically relevant contribution. 

We look forward to receiving your revised manuscript.

Kind regards,

Prof. Paola Gremigni, Ph.D.

Academic Editor

PLOS ONE

Journal Requirements:

3. We note that Table 6/other information may include questionnaire items that may have been previously published. The reproduction of previously published work has implications for the copyright that may apply to these publications. We would be grateful if you could clarify whether you have obtained permission from the original copyright holder to republish these items under a CC BY license. If you have not obtained permission to publish these items please remove them from your manuscript. You may wish to replace the text you have removed with relevant question numbers/ brief descriptions of each item; please be sure to include any relevant references and in-text citations.

Reviewers' comments:

Reviewer's Responses to Questions

**Comments to the Author**

1. Is the manuscript technically sound, and do the data support the conclusions?

Reviewer #1: Yes

Reviewer #2: Yes

2. Has the statistical analysis been performed appropriately and rigorously? 

Reviewer #1: Yes

Reviewer #2: Yes

3. Have the authors made all data underlying the findings in their manuscript fully available?

Reviewer #1: Yes

Reviewer #2: Yes

4. Is the manuscript presented in an intelligible fashion and written in standard English?

Reviewer #1: Yes

Reviewer #2: Yes

5. Review Comments to the Author

Reviewer #1: The manuscript (Confirmatory factor analysis of the EORTC QLQ-C30 and BR45 to develop the Breast Utility Instrument, a preference-based instrument to measure health related quality of life in women with breast cancer) is well organized and written clearly in Standard English. Also, the methodology is described clearly, and presented results seem to support the authors’ conclusions.

Based on above comment, It is suggested that:

1. Mention the validity and reliability of the tools used in the research

2. In the first paragraph of the results, it is explained about the participants, which seems to be related to the research procedure, and it is better to mention this part in procedure.

3. The exclusion criteria are not explicitly explained.

4. The treatment recieved by patients is a bit ambiguous and it is not clear whether all of them received hormonal treatment or benefited from other treatments.

Reviewer #2: This is a good study with very finely research design. The findings have research and clinical values for patients with breast cancer. However, there are some errors in numbers in table 1. Please check the correctness of numbers in all tables.

6. PLOS authors have the option to publish the peer review history of their article (what does this mean?). If published, this will include your full peer review and any attached files.

Reviewer #1: No

Reviewer #2: No

---

## [Author Response · Author response to Decision Letter 0]

8 Nov 2021

Developing the Breast Utility Instrument, a preference-based instrument to measure health related quality of life in women with breast cancer: confirmatory factor analysis of the EORTC QLQ-C30 and BR45 to establish dimensions 

(PONE-D-21-27345)

Response to Reviewers

Thank you for taking an interest in our study and for providing helpful comments and suggestions. We have considered all your comments – please find our responses below. The corrections by line number refer to the Manuscript file.

Academic Editor’s comments:

My personal opinion is that the manuscript is not easy to follow by readers and researchers who are not strictly familiar with statistical and psychometric issues. Therefore, I would appreciate a tentative of explaining better some technical aspects.

Reply:

Thank you for your comment. We have made edits to the following sections in the manuscript to improve our explanation of the technical aspects.

Page 9, lines: 185-189 explain why preliminary analyses are performed:

“We conducted preliminary analyses prior to the CFA. First, the item response distributions of the QLQ-C30 and BR45 over BrC health states were visualized using stacked bar plots. Next, correlations were inspected: inter-item, item-to-dimension (subscale), and inter-dimensional (S3 Figure), to ensure there was a sufficient association between items and dimensions to move forward with CFA.”

Page 11, lines 224-226 explain the concept of global model fit:

“Global model fit is a descriptive indicator of how well the model reproduces the observed relationships between the indicators, represented by items, in the input matrix (31). We used five tests to evaluate global model fit:” 

Page 12, lines 242-244 explain saliency:

“Saliency evaluates if the items are associated with the pre-specified factor. We inspected factor loadings (λ) and considered items with λ>0.3 with statistical significance at � = 0.01 (with Bonferroni correction) to be salient to a given factor (36).”

Page 12, lines 249-254 explain the modification index:

“A modification index approximates the degree that a model’s χ2 statistic would decrease if a given fixed parameter became freely estimated, analogous to the χ2 difference (with a single degree of freedom) of nested models (31). Therefore, well-fitted models have small modification indices. A model with local areas of strain would have item pairs in the same dimension with high residuals (>0.4), and high modification indices (>25) (40).”

Page 12, lines 254-256 explain the rationale for applying residual correlations:

“If there was substantial clinical rationale and overlapping item content, we re-specified models to correlate item residuals, and re-assessed residuals and modification indices.”

Journal Requirements

Reply:

Thank you for noticing these discrepancies – our apologies for these deviations from PLOS ONE’s style requirements. 

Each Figure is now saved as a separate file according to your file name requirements. The caption of each figure now appears directly after the paragraph in which it was first cited.

Each Table is placed in the manuscript directly after the paragraph in which it was first cited. 

Reply:

Thank you for this note to review our reference list. Here are the changes to our reference list, with reasons provided: 

We removed this reference as it is still awaiting publication:

 Tsui TCO, Torres S, Bielecki J, Davis AM, Mitsakakis N, Trudeau M, et al. Creating a map for the road less travelled: A scoping review and framework of developing condition-specific preference-based instruments In submission. 2021.

We replaced our initial reference for the World Health Organization’s quality of life instrument (WHOQoL) instrument: 

 The World Health Organization Quality of Life Assessment (WHOQOL): development and general psychometric properties. Soc Sci Med. 1998;46(12):1569-85.

This reference has been replaced with a more appropriate reference of WHO’s definition of health:

 Constitution of the World Health Organization: World Health Organization; 2006. Available from: https://www.who.int/governance/eb/who_constitution_en.pdf.

We also updated wording to reflect WHO’s core health dimensions on page 8, line 164, and page 32, line 435.

To address the reviewer’s comments about elaborating on the measurement properties of the EORTC QLQ-C30 and BR45, the following references have been added:

 Osoba D, Zee B, Pater J, Warr D, Kaizer L, Latreille J. Psychometric properties and responsiveness of the EORTC quality of Life Questionnaire (QLQ-C30) in patients with breast, ovarian and lung cancer. Qual Life Res. 1994;3(5):353-64.

 Groenvold M, Klee MC, Sprangers MA, Aaronson NK. Validation of the EORTC QLQ-C30 quality of life questionnaire through combined qualitative and quantitative assessment of patient-observer agreement. J Clin Epidemiol. 1997;50(4):441-50.

All our references are current based on the best of our knowledge. No papers have been retracted.

3. We note that Table 6/other information may include questionnaire items that may have been previously published. The reproduction of previously published work has implications for the copyright that may apply to these publications. We would be grateful if you could clarify whether you have obtained permission from the original copyright holder to republish these items under a CC BY license. If you have not obtained permission to publish these items please remove them from your manuscript. You may wish to replace the text you have removed with relevant question numbers/ brief descriptions of each item; please be sure to include any relevant references and in-text citations. 

Reply:

Thank you for noticing the questionnaire items in Table 6. The wording in Table 6 has been modified to include the question number and a brief descriptor of the item as you have kindly suggested. 

Reply:

Thank you for noting this discrepancy. We have listed the THETA Fund for Excellence (# 5790 6839 0706) and Dr. Kathleen Pritchard’s support in both sections. Dr. Pritchard provided a direct donation to the Biomatrix, therefore that funding source does not have a grant number.

b) If there are no restrictions, please upload the minimal anonymized data set necessary to replicate your study findings as either Supporting Information files or to a stable, public repository and provide us with the relevant URLs, DOIs, or accession numbers. For a list of acceptable repositories, please see http://journals.plos.org/plosone/s/data-availability#loc- recommended-repositories. 

Thank you for your clarification. 

Reply:

The following clarification has been added to the revised cover letter: 

Data cannot be shared publicly because data access is restricted by patient consent to use their study data. Anonymized data are available and can be obtained with approval of the Chairs of the Sunnybrook Research Institute Research Ethics Board (https://sunnybrook.ca/research/content/?page=sri-crs-reo-home) and the University Health Network Research Ethics Board (http://www.uhnresearch.ca/service/responsible-conduct), and approved institutional data sharing agreements. In Canada, research data are the property of the institution, not the investigators. Request for the data may be sent to the corresponding author (teresa.tsui@utoronto.ca).

5. Review Comments to the Author 

Reviewer #1: The manuscript (Confirmatory factor analysis of the EORTC QLQ-C30 and BR45 to develop the Breast Utility Instrument, a preference-based instrument to measure health related quality of life in women with breast cancer) is well organized and written clearly in Standard English. Also, the methodology is described clearly, and presented results seem to support the authors’ conclusions. 

Based on above comment, It is suggested that:

1. Mention the validity and reliability of the tools used in the research 

Reply: 

Please see page 5 lines 94-105 and page 6 lines 114-118 for the validity and reliability of the QLQ C30 and BR45, respectively.

Page 5, lines 94-105, QLQ C30

“The QLQ C30 has demonstrated measurement properties in a range of cancers including breast cancer (16). It has an established factor structure (construct validity) consistent with the original development population in lung cancer and internal consistency (Cronbach’s alpha >0.7 for all subscales except for role functioning and cognitive functioning where alpha was <0.70). Discrimination between local-regional and metastatic BrC was demonstrated in 6/9 subscales at pre-treatment (p<0.002) and in 4/9 subscales (p<0.002) 8 days after chemotherapy (16). Comparing local-regional and metastatic BrC, subscales without significant difference in mean scores pre-treatment were: emotional functioning, cognitive functioning, and nausea / vomiting, and subscales without significant differences 8 days after chemotherapy were: emotional functioning, social functioning, cognitive functioning, nausea and vomiting, and fatigue (16). The QLQ C30 has established patient-observer agreement with a median kappa = 0.5 (range: 0.49-1.00) in patients with breast and gynecological cancers (17).”

Page 6, lines 114-118, BR45

“The developers of the BR45 pre-tested the breast module to evaluate the importance, comprehensibility, and acceptability of its questionnaire items (face validity and feasibility) (14). The BR45 has also established preliminary psychometric properties, where all subscales have acceptable internal consistency (Cronbach's alpha > 0.7), and the three new symptom subscales and new satisfaction subscale had no strong correlation with the existing BR23 subscales (14).” 

2. In the first paragraph of the results, it is explained about the participants, which seems to be related to the research procedure, and it is better to mention this part in procedure. 

Reply:

Thank you for this suggestion. We moved the first paragraph of the results to the first paragraph under participants and procedures, patients, integrating the text with the original text. This is presented on page 6, in lines 122-173.

Our clinician expert recommended that we clarify the recurrence state (R), as local recurrence, to distinguish recurrence from distant recurrence (metastatic disease). We therefore added local before to all instances of recurrence – lines 136, 137, 138, 142, 143, 301.

3. The exclusion criteria are not explicitly explained. 

Reply:

Thank you – please see edited sentence where we explicitly explain the exclusion criteria, now page 7, lines 131-132:

“Patients were excluded if they had non-invasive BrC, anther primary cancer within the prior five years, or did not understand English and did not have a translator.”

4. The treatment received by patients is a bit ambiguous and it is not clear whether all of them received hormonal treatment or benefited from other treatments. 

Reply:

Thank you for pointing this out. we have elaborated on the treatment intents received by patients on page 14, lines 357-359. 

“The most common treatment intents were adjuvant (64.2%), palliative (22.5%), and neoadjuvant (6.4%). The most common treatment regimens were endocrine therapy (57.0%), chemotherapy (17.1%), and targeted therapy (16.7%).”

More detailed treatments are also presented in Table 1, under the sub-headings Surgery; Surgery – axillary; Intent of systemic therapy; Regimen; Chemotherapy; Targeted therapy; Endocrine therapy; Other - bone modifying agents; Radiotherapy intent.

Reviewer #2: This is a good study with very finely research design. The findings have research and clinical values for patients with breast cancer. However, there are some errors in numbers in table 1. Please check the correctness of numbers in all tables. 

Reply:

Thank you for this feedback and for the opportunity to improve our Table 1. We have checked the accuracy of the numbers in Table 1 and noticed that we initially mis-spelled the Charlson Comorbidity Index (CCI). We now present the CCI as a categorical variable. 

The population comparators and corresponding population references have been clarified. We corrected some typographical errors under population comparators. We also reformatted Table 1 to simplify the original layout. Table 1 now starts on page 14, line 289.

---

## [Editor Report · Decision Letter 1]

3 Jan 2022

Developing the Breast Utility Instrument, a preference-based instrument to measure health-related quality of life in women with breast cancer: confirmatory factor analysis of the EORTC QLQ-C30 and BR45 to establish dimensions

PONE-D-21-27345R1

Dear Dr. Tsui,

We’re pleased to inform you that your manuscript has been judged scientifically suitable for publication and will be formally accepted for publication once it meets all outstanding technical requirements.

Kind regards,

Paola Gremigni, Ph.D.

Academic Editor

PLOS ONE

Additional Editor Comments (optional):

Dear Authors,

I appreciated your appropriate answers to all the Reviewers' and this Editor's observations.
---

## [Editor Report · Acceptance letter]

27 Jan 2022

PONE-D-21-27345R1 

Developing the Breast Utility Instrument, a preference-based instrument to measure health-related quality of life in women with breast cancer: confirmatory factor analysis of the EORTC QLQ-C30 and BR45 to establish dimensions 

Dear Dr. Tsui:

I'm pleased to inform you that your manuscript has been deemed suitable for publication in PLOS ONE. Congratulations! Your manuscript is now with our production department. 

Kind regards, 

on behalf of

Prof. Paola Gremigni 

Academic Editor

PLOS ONE